# Achievement of High Perpendicular Anisotropy and Modification of Heat Treatment Peeling in Micron-Thickness Nd-Fe-B Films Used for Magnetic MEMS

**DOI:** 10.3390/ma16114071

**Published:** 2023-05-30

**Authors:** Jingbin Huang, Zhanyong Wang, Shijie Liao, Fang Wang, Min Huang, Jian Zhang

**Affiliations:** 1CAS Key Laboratory of Magnetic Materials and Devices, Ningbo Institute of Materials Technology and Engineering, Chinese Academy of Sciences, Ningbo 315201, China; huangjingbin@nimte.ac.cn (J.H.); liaoshijie@nimte.ac.cn (S.L.); 2Center of Materials Science and Optoelectronics Engineering, University of Chinese Academy of Sciences, Beijing 100049, China; 3Zhejiang Province Key Laboratory of Magnetic Materials and Application Technology, Ningbo Institute of Materials Technology and Engineering, Chinese Academy of Sciences, Ningbo 315201, China; 4School of Materials Science and Engineering, Shanghai Institute of Technology, Shanghai 201418, China; wzy2195@sit.edu.cn; 5School of Materials and Chemical Engineering, Ningbo University of Technology, Ningbo 315211, China; wangfangch@nbut.edu.cn

**Keywords:** Nd-Fe-B film, out-of-plane anisotropy, magntic MEMS

## Abstract

Thick Nd-Fe-B permanent magnetic films with good perpendicular anisotropy have important applications in magnetic microelectromechanical systems (MEMSs). However, when the thickness of the Nd-Fe-B film reaches the micron level, the magnetic anisotropy and texture of NdFeB film will become worse, and it is also prone to peeling during heat treatment, which seriously limits their applications. In this paper, Si(100)/Ta(100 nm)/Nd_x_Fe_91−x_B_9_(x = 14.5, 16.4, 18.2)/Ta (100 nm) films with thicknesses of 2–10 μm are prepared by magnetron sputtering. It is found that gradient annealing (GN) could help improve the magnetic anisotropy and texture of the micron-thickness film. When the Nd-Fe-B film thickness increases from 2 μm to 9 μm, its magnetic anisotropy and texture do not deteriorate. For the 9 μm Nd-Fe-B film, a high coercivity of 20.26 kOe and high magnetic anisotropy (remanence ratio M_r_/M_s_ = 0.91) are achieved. An in-depth analysis of the elemental composition of the film along the thickness direction is conducted, and the presence of Nd aggregation layers at the interface between the Nd-Fe-B and the Ta layers is confirmed. The influence of thicknesses of the Ta buffer layer on the peeling of Nd-Fe-B micron-thickness films after high-temperature annealing is investigated, and it is found that increasing the thickness of the Ta buffer layer could effectively inhibit the peeling of Nd-Fe-B films. Our finding provides an effective way to modify the heat treatment peeling of Nd-Fe-B films. Our results are important for the development of Nd-Fe-B micron-scale films with high perpendicular anisotropy for applications in magnetic MEMS.

## 1. Introduction

Permanent magnets are used in magnetic microelectromechanical systems (MEMSs) through electrostatic, electromagnetic, and other effects, typically ranging in thickness from a few microns to a few millimeters. High-performance permanent magnetic films are the key to implementing magnetic MEMS components [1,2,3]. The Nd-Fe-B thick film with perpendicular anisotropy deposited on semiconductor Si substrate is compatible with microelectronics manufacturing technology, and has important applications in micro-components such as micropumps, microvalves, micromotors, brakes, and energy harvesters [4,5,6]. For high-performance NdFeB thin films used in magnetic microelectromechanical systems (MEMSs), not only does the thickness need to reach the micrometer level (>1 μm) [7], but it is also necessary for the film to have magnetic anisotropy perpendicular to the film plane [8,9,10]. However, there seems to be a certain contradiction between the achievements of high perpendicular anisotropy and large thickness. With the increase in the thickness of NdFeB film (>1 μm), the magnetic anisotropy of the film deteriorates and tends to form magnetic isotropy. It is still desirable to improve the magnetic properties of micron-sized Nd-Fe-B films.

Dempsey et al. reported the fabrication of 5 μm thick NdFeB films with strong perpendicular anisotropy on Si wafer ((BH)_max_ = 400 kJ/m^3^, μ_0_H_c_ = 1.6 T, and μ_0_Mr = 1.4 T) [11]. Its excellent magnetic perpendicular anisotropy comes from the columnar microstructure of the Nd-Fe-B layer, with the c-axis of the Nd_2_Fe_14_B grain being oriented along the perpendicular direction of the substrate surface. Further studies have found that high-performance NdFeB films are closely related to the microstructure composed of the Nd_2_Fe_14_B main phase surrounded by the non-magnetic Nd-rich grain boundary phase, and the refined Nd_2_Fe_14_B grains and uniform Nd-rich thin boundary phase are important for the improvement in coercivity [12]. Usually, when the thickness of NdFeB films is higher than 5 μm, their magnetic anisotropy and texture will become worse [13,14,15,16,17]. It is of great significance to investigate how to prepare thick NdFeB films with high perpendicular magnetic anisotropy. The formation of textured NdFeB film is affected by the deposition process (deposition temperature, sputtering pressure, and power) [18,19] and film composition. The one-step crystallization by thermal deposition or annealing after thermal deposition is conducive to the formation of perpendicular magnetic anisotropy [20,21,22,23,24]. In addition, buffer materials with (110) orientation (Ta, Mo, Cr, et al.) also have a guiding effect on the c-axis orientation of NdFeB films [25,26,27,28].

The micrometer-thickness NdFeB films have the problem of the film peeling off during high-temperature annealing, which greatly limits their application. The peeling of NdFeB films is related to the large difference in thermal expansion between the film and the Si substrate. During thermal annealing, the thermal stress on the film exceeds the critical cracking stress of the film, making it easy to peel off the substrate. With the increase in film thickness, the thermal stress of the film during the thermal annealing process also increases, and the tendency of film peeling also increases. Studies have shown that the thermal stress on the film peeling during thermal annealing can be weakened by etching a mask trench on the Si substrate and then depositing the NdFeB film [11,29]. However, this process makes film fabrication more complicated.

In this paper, the micron-sized (1–10 μm) NdFeB films are prepared by composite alloy targets with different Nd content, and the effects of preparation conditions and film thickness on film composition, texture, and magnetic properties are investigated. It is found that annealing with a temperature gradient could help to improve the texture of thick films. A 9 μm thick NdFeB film with high coercivity H_c_ = 20.26 kOe and very good out-of-plane anisotropy is achieved. In addition, the effect of Ta buffer layer thickness on the tolerance of film to annealing thermal stress is explored, and it was found that increasing the Ta buffer thickness can effectively prevent the peeling of the NdFeB film from the substrate during the high-temperature annealing.

## 2. Materials and Methods

Si(100)/Ta/Nd-Fe-B(2–9 μm)/Ta films are fabricated by magnetron sputtering. The base vacuum pressure of magnetron sputtering machine is better than 3 × 10^−6^ Pa. A commercial Ta target (99.99%) is used for preparing the buffer layer and capping layer, and the deposition of Ta is completed under the conditions of a deposition temperature of 300 °C, a DC sputtering power of 120 W, and an Ar pressure of 0.8 Pa, respectively. The composition of the 60 mm diameter Nd-Fe-B composite targets is analyzed by inductively coupled plasma emission spectrometer (ICP, SPECTRO ARCOS, SPECTRO, Kleve, Germany), and their compositions are Nd_14.5_Fe_76.5_B_9_, Nd_16.4_Fe_74.6_B_9_, Nd_18.2_Fe_72.8_B_9_, and Nd_14.5_Fe_74.5_B_11_. When the substrate temperature reaches the deposition temperature, the Nd-Fe-B film is deposited, and the sputtering power, Ar pressure, and Ar flow rate are 100–140 W, 0.8–1.2 Pa, and 60 cm^3^ min^−1^, respectively. After deposition, the film is crystallized by in situ temperature gradient annealing (GN) or traditional ex situ annealing without temperature gradient (NA). An X-ray diffractometer (XRD-D8 ADVANCE, Bruker, Karlsruhe, Germany) with Cu Kα radiation is used to detect the crystal structure and texture of the film. The hysteresis loops are measured by the physical property measurement system (PPMS, Quantum Design company, San Diego, CA, USA). The microstructure and elemental distribution of the cross-section and surface of the film are observed by scanning electron microscope (SEM, FEI Quanta FEG 250, FEI, Oregon, OR, USA) with an energy-dispersive X-ray spectrometer (EDS). Glow discharge emission spectrometer (GDA 750HP, Spectruma Analytik GmbH, Frankfurt, Germany) is used to analyze the distribution of the elements in the film thickness direction.

## 3. Results and Discussion

### 3.1. Phase Structure and Magnetic Properties of Micron-Thickness Nd-Fe-B Films

The content of Nd in Nd-Fe-B films is essential for the achievement of high performance in NdFeB films. We prepared micron-thickness NdFeB films with different Nd content using Nd_x_Fe_91−x_B_9_ (x = 14.5, 16.4, 18.2) targets. Different heat treatment processes may have a great influence on the texture and magnetic anisotropy of the film. We investigated the effects of GN and NA on the magnetic anisotropy of micrometer thickness Nd-Fe-B films. Figure 1 shows the hysteresis loop of Si(100)/Ta(100 nm)/Nd_x_Fe_91−x_B_9_ (x = 14.5, 16.4, 18.2) (2 μm)/Ta(100 nm) films. By comparing the hysteresis loops of the film with x = 14.5 after GN and NA heat treatment, it is found that the film after GN heat treatment has a higher remanence ratio (M_r_/M_s_) and better magnetic anisotropy. This manifests that GN heat treatment can effectively improve the magnetic anisotropy of micron-thickness Nd-Fe-B films. The measurement results for the hysteresis loops of films with different Nd content after GN heat treatment indicate that with the increase in Nd content, the coercivity of the film increases, while the remanence (M_r_) and M_r_/M_s_ (M_s_ is saturation magnetization) drop. The reduction of M_r_/M_s_ indicates that the out-of-plane anisotropy of the film becomes worse. When the Nd content increases from x = 16.4 to x = 18.2, the film coercivity (H_c_) increases slightly, while M_r_ decreases sharply. Nd-Fe-B materials with high coercivity typically require a Nd content that exceeds the Nd_2_Fe_14_B ratio. Under 140 Wsputtering power, as the Nd content of the target increases, the Nd content in the film also increases, resulting in an increase in the coercivity. The results in Figure 1 also reveal that too much Nd content in the film is detrimental to the magnetic properties. For the film with x = 14.5, the H_c_, M_r_, M_r_/M_s_, and (BH)_max_ (maximum energy product) are 10.15 kOe, 12.47 kGs, and 0.96 and 38.9 MGOe, respectively. It has a high magnetic anisotropy but low coercivity. For the film with x = 16.4, the H_c_, M_r_, M_r_/M_s_, and (BH)_max_ are 18.45 kOe, 9.61 kGs, 0.81, and 22.60 MGOe, respectively. Compared with the films with x = 14.5 and x = 18.2, the films with x = 16.4 has better comprehensive magnetic properties.

The phase structure of Si(100)/Ta(100 nm)/Nd_x_Fe_91−x_B_9_(x = 14.5, 16.4, 18.2) (2 μm)/Ta(100 nm) films was characterized by X-ray diffraction (XRD) (Figure 2). The unannealed Nd-Fe-B film detected only Si peaks and Ta peaks but no diffraction peaks of the Nd_2_Fe_14_B phase, indicating that the as-deposited NdFeB film was in an amorphous state. The XRD patterns of annealed films show peaks of Nd_2_Fe_14_B phase (P42/mnm (136); JCPDS, 89-3632) and Nd-rich phase (P63/mmc (194); JCPDS, 65-3424). With the increase in Nd content, the peak intensity of the Nd-rich phase increases. The film with x = 14.5 has strong (004), (105), (006), and (008) diffraction peaks, indicating that it possesses a strong texture, and the c-axis of Nd_2_Fe_14_B grains is orientated along the out-of-plane of the film surface. With the increase in Nd content in films (x = 16.4, 18.2), the diffraction peaks of (410), (214), (115), and (116) appear, indicating a decrease in the c-axis orientation of Nd-Fe-B grains. The results above are consistent with the magnetic properties shown in Figure 1.

The effects of thickness on the structure and magnetic properties of micron-thickness NdFeB films are further investigated. Ta(100 nm)/Nd_16.4_Fe_74.6_B_9_/Ta(100 nm) films with a thickness of 2 μm, 5 μm, and 9 μm were prepared under the conditions of sputtering power of 140 W, deposition temperature of 550 °C and GN heat-treatment of 750 °C × 20 min. Figure 3a–c show the hysteresis loops for NdFeB films of different thicknesses measured along directions parallel and perpendicular to the film surface. The H_c_ and M_r_/M_s_ of 2 μm, 5 μm, and 9 μm thickness films are 18.45 kOe/15.70 kOe/20.26 kOe and 0.81/0.85/0.91, respectively. The high coercivities are achieved for films of 2–9 μm. Usually, as the thickness of the film increases, its remanence ratio and magnetic anisotropy will drop. However, the magnetic anisotropy of the micro-thickness film we prepared does not deteriorate as the thickness grows up. The film with a thickness of 9 μm still has a very good magnetic anisotropy, which should be attributed to the heat treatment using gradient annealing. For the same thickness 9 μm films, the film annealed at 800 °C can achieve a higher remanence ratio (magnetic anisotropy) than that annealed at 750 °C, which may also be related to GN. The higher the temperature, the greater the temperature gradient difference generated, which can enhance the magnetic anisotropy of the film. Compared with the film annealed at 750 °C, the film annealed at 800 °C has improved magnetic energy product owing to the increased magnetic anisotropy, although the coercivity of the latter film is slightly reduced.

Figure 4 presents the XRD diffraction patterns of Ta (100 nm)/Nd_16.4_Fe_74.6_B_9_/Ta (100 nm) films with different thicknesses. All the films exhibit characteristic peaks (004), (105), (006), and (008), indicating the c-axis of Nd-Fe-B grains orientates along the perpendicular direction of the film plane. As the film thickness increases, these orientation characteristic peaks do not weaken, but strengthen. These results are consistent with the magnetic properties shown in Figure 2. It further confirms that the thick Nd-Fe-B film (9 μm) annealed with GN can indeed achieve good out-of-plane magnetic anisotropy.

### 3.2. Composition Analysis and Micromorphology of Micron Thickness NdFeB Films

The composition of the film has an important influence on its properties. Generally, the composition analysis along the film thickness direction is rarely characterized. We analyze in-depth the elemental composition of the film along the thickness direction by glow discharge emission spectrometer. Figure 5a shows the distribution of elements for the Ta(100 nm)/Nd_14.5_Fe_76.5_B_9_ (2 μm)/Ta(100 nm) film fabricated with sputtering power 140 W, deposition temperature 500 °C, and 700 °C × 20 min (GN). It shows that Nd is uniformly distributed in the Nd-Fe-B layer, but the aggregation of Nd appears at the interface between the Nd-Fe-B and Ta layers. This may be related to the extrusion of Nd-Fe-B by the Ta layer during the high-temperature deposition [30,31], which makes Nd easy to aggregate at the interface. The composition of the Nd-Fe-B layer is analyzed, and the nominal composition obtained is Nd_14.1_Fe_74.0_B_11.9_ (at. %), which is not much different from the composition of the target Nd_14.5_Fe_76.5_B_9_ (at. %). Figure 5b presents the composition distribution of a 100 W sputtered, 550 °C deposited and unannealed Ta(100 nm)/Nd_14.5_Fe_74.5_B_11_ (2 μm)/Ta(100 nm) film along the thickness direction. It can be seen that this unannealed film also accumulates higher levels of Nd at the interface between the Nd-Fe-B and Ta buffer or overlay layer, which should be due to the thermal deposition process. In the Nd-Fe-B layer, the Nd content undulates in the thickness direction. This may be related to the deposition of our films in multiple time periods. For the initial sputtering and ensuing stable sputtering for a period of time, the amount of Nd deposited into the film is different. The Ar^+^ bombardment will heat up the target, which may affect the sputtering yield of Nd and also the content of Nd in the film. With the extension of time, the target temperature tends to stabilize, and the Nd content in the film also reaches stability. The composition of the Nd-Fe-B layer is analyzed, and it is Nd_11.2_Fe_74.0_B_14.8_. The Nd content in 100 W sputtered Nd-Fe-B film is lower than that in 140 W sputtered film, which is in agreement with the previously reported results [32].

Figure 6a is the SEM cross-sectional image of Ta(100 nm)/Nd_16.4_Fe_74.6_B_9_(9 μm)/Ta(100 nm) films with GN heat treatment at 800 °C for 20 min. The thickness of the Nd-Fe-B layer can be determined to be about nine microns. Nd_2_Fe_14_B grains form a columnar crystal, indicating that they have a preferential orientation characteristic. This further supports the existence of texture and magnetic anisotropy in the film. Figure 6b shows the film surface morphology taken by SEM. Ridged protrusions distributed in a network manner appear on the surface of the film. It has been reported that during the thermal annealing process, the compressive stress on the Nd-Fe-B layer can be generated by the Ta capping layer, and the Nd-rich liquid phase is subjected to compressive stress and squeezed on the surface, resulting in the formation of a ridge-like structure in the Ta surface [30]. This extrusion is conducive to achieving uniform coating of Nd_2_Fe_14_B grains by the thin Nd-rich phase and obtaining high coercivity [30]. Figure 6c shows the elements distribution of EDS on the surface of the film, proving the ridged protrusions are the Nd-rich phase.

### 3.3. The Thickening of Buffer Layer Inhibits the Peeling of NdFeB Films

Due to the different coefficients of thermal expansion for Nd-Fe-B film and Si substrate, the thick Nd-Fe-B films easily peel off from the substrates during high-temperature heat treatment. This seriously limits the preparation and application of Nd-Fe-B thick films. To address this issue, we investigated the effect of Ta buffer layer thicknesses on the peeling of high-temperature heat treated Nd-Fe-B micron thickness films. Figure 7 and Figure 8 are SEM surface morphology images of Ta (x nm)/Nd_14.5_Fe_74.5_B_11_ (6 μm)/Ta(100 nm) (x = 100, 230, 380, 500, 650, 1000) film with NA heat treatment at 850 °C × 20 min and 900 °C × 20 min, respectively. Figure 9 summarizes the peeling behavior of NdFeB films with different Ta buffer layer thicknesses at different annealing temperatures. When the Ta buffer layer thickness is 100 nm, cracks appear on the surface of the film after annealing at 850 °C, and the cracks are interconnected. This indicates that Ta with a thickness of 100 nm is not enough to buffer the thermal stress generated under annealing at 850 °C. When the thickness of the Ta buffer layer increases to over 230 nm, no cracks appear in the film, indicating that increasing the thickness of the Ta buffer layer is beneficial for inhibiting the peeling of the film. For Nd-Fe-B films heat-treated at 900 °C, films with Ta buffer layer thickness below 380 nm all exhibited large-scale delamination. When the thickness of the Ta buffer layer increases to 500 nm, the capping layer, along with the magnetic Nd-Fe-B layer, is partially peeled off. Our experimental results above indicate that a certain thickness of the Ta buffer layer can buffer the thermal stress caused by heat treatment and effectively inhibit the peeling of micron-thickness Nd-Fe-B films. As the annealing temperature increases, it is necessary to increase the thickness of the buffer layer to suppress the detachment of Nd-Fe-B films. Our results provide an effective way to suppress film detachment during high-temperature heat treatment.

## 4. Conclusions

The Si(100)/Ta(100 nm)/Nd_x_Fe_91−x_B_9_(x = 14.5, 16.4, 18.2)/Ta(100 nm) films with thicknesses of 2–10 μm are prepared by magnetron sputtering. It is found that gradient annealing helps to improve the magnetic anisotropy and out-of-plane texture of thick Nd-Fe-B films. The influences of Nd content and film thickness on the texture and magnetic properties of Nd-Fe-B films heat-treated with GN are investigated. As the Nd content increases, the coercivity of the film increases, while the magnetic anisotropy drops. It is noteworthy that with the increase in the film thickness from 2 μm to 9 μm, the magnetic anisotropy and out-of-plane texture of the film do not deteriorate. For Ta(100 nm)/Nd_16.4_Fe_74.6_B_9_ (9 μm)/Ta(100 nm) film, the high coercivity of 20.26 kOe and high magnetic anisotropy (M_r_/M_s_ = 0.91) are achieved. The film composition along the thickness direction is analyzed, and an Nd-rich phase layer between the Nd-Fe-B and Ta layers is confirmed. The microstructure of micron-thickness film shows the columnar grains perpendicular to the film surface. Increasing the thickness of the Ta buffer layer is beneficial for inhibiting the peeling of micrometer-scale films during high-temperature annealing. A 230 nm Ta buffer layer can buffer the thermal stress in film generated by 850 °C annealing, while a 650 nm Ta buffer layer can buffer the thermal stress generated by 900 °C annealing. Our results provide important routes to improve the magnetic anisotropy and prevent the peeling of micron-thickness Nd-Fe-B films during high-temperature treatment, which is of great significance for their applications in magnetic MEMSs. Micromagnetic components made of high perpendicular anisotropic Nd-Fe-B permanent magnet films integrated into semiconductor Si can be applied to micromotors, switches, brakes, energy harvesters, etc. Our findings contribute to the development of these devices.

## Figures and Tables

**Figure 1 materials-16-04071-f001:**
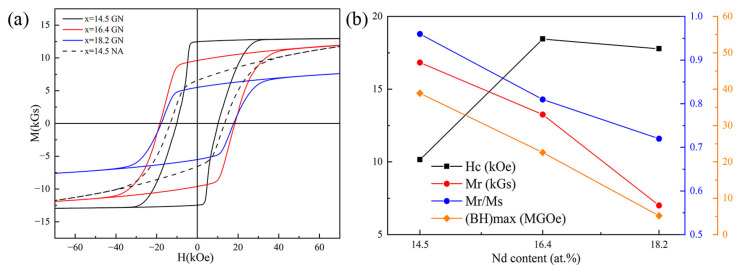
Si(100)/Ta(100 nm)/Nd_x_Fe_91−x_B_9_(x = 14.5, 16.4, 18.2) (2 μm)/Ta(100 nm) films fabricated with a sputtering power of 140 W. (**a**) Hysteresis loop; (**b**) the relationship between the Nd content and magnetic properties of the films after GN heat treatment. (x = 14.5 GN: Ta(100 nm)/Nd_14.5_Fe_76.4_B_9_ (2 μm)/Ta(100 nm) deposited at 500 °C, annealed at 700 °C × 20 min (GN); x = 16.4 GN: Ta(100 nm)/Nd_16.4_Fe_74.6_B_9_ (2 μm)/Ta(100 nm) deposited at 550 °C, annealed at 750 °C × 20 min (GN); x = 18.2 GN: Ta(100 nm)/Nd_18.2_Fe_72.8_B_9_ (2 μm)/Ta(100 nm) deposited at 550 °C, annealed at 700 °C × 20 min (GN); and x = 14.5 NA: Ta(100 nm)/Nd_14.5_Fe_74.5_B_11_/Ta(100 nm) films deposited at 550 °C, annealed at 700 °C × 10 min (NA)).

**Figure 2 materials-16-04071-f002:**
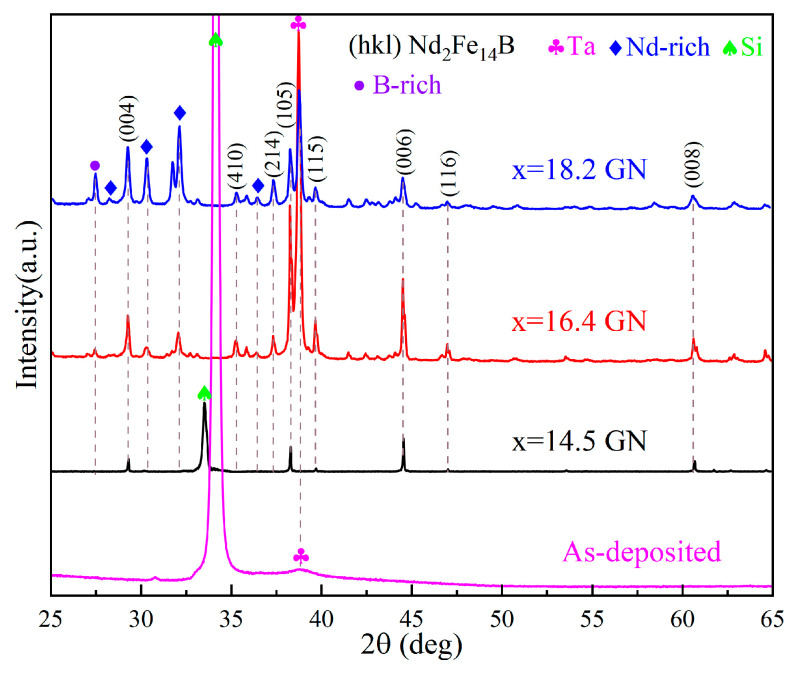
The XRD patterns of Si(100)/Ta(100 nm)/Nd_x_Fe_91−x_B_9_(x = 14.5, 16.4, 18.2) (2 μm)/Ta(100 nm) films shown in Figure 1. (The dotted lines connect the XRD peak patterns at the same 2θ angle).

**Figure 3 materials-16-04071-f003:**
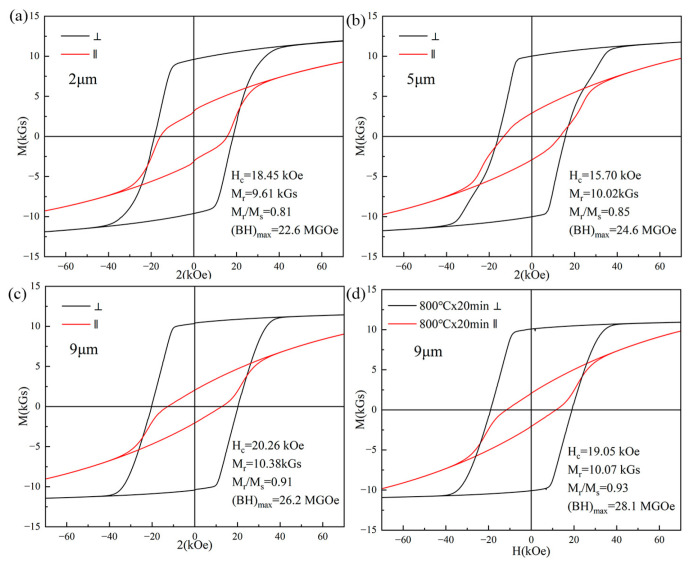
Hysteresis loops of Ta(100 nm)/Nd_16.4_Fe_74.6_B_9_/Ta(100 nm) films of different thicknesses measured along the directions parallel (‖) and perpendicular (⊥) to the film surface. (**a**) The 2 μm film with 750 °C × 20 min annealing (GN); (**b**) 5 μm film with 750 °C × 20 min annealing (GN); (**c**) 9 μm film with 750 °C × 20 min annealing (GN); and (**d**) 9 μm film with 800 °C × 20 min annealing (GN).

**Figure 4 materials-16-04071-f004:**
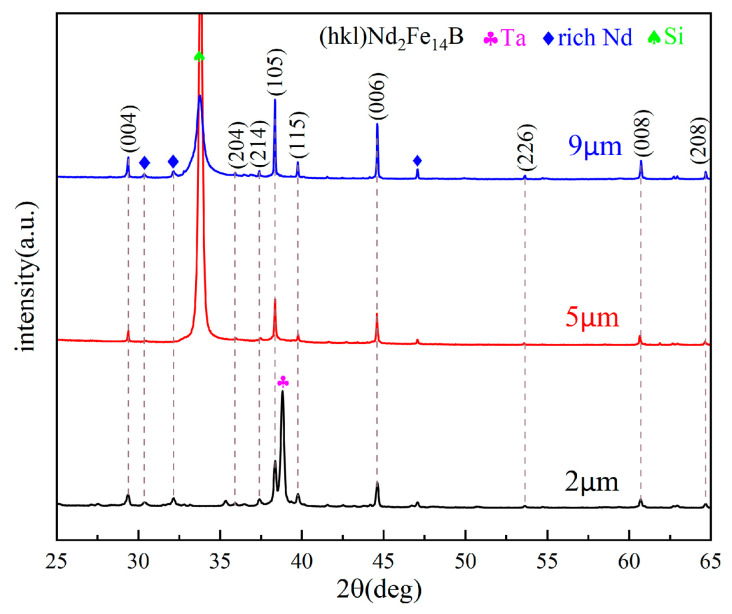
The XRD patterns of Si(100)/Ta(100 nm)/Nd_16.4_Fe_74.6_B_9_/Ta(100 nm) films for different thicknesses after 750 °C × 20 min annealing (GN). (The dotted lines connect the XRD peak patterns at the same 2θ angle).

**Figure 5 materials-16-04071-f005:**
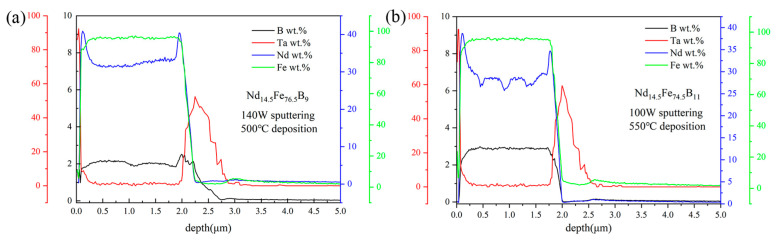
The dependence of composition on film thickness for (**a**) Ta(100 nm)/Nd_14.5_Fe_76.5_B_9_(2 μm)/Ta(100 nm)/Ta(100 nm) film after GN treatment at 700 °C × 20 min and (**b**) Ta(100 nm)/Nd_14.5_Fe_74.5_B_11_(2 μm)/Ta(100 nm) film without annealing.

**Figure 6 materials-16-04071-f006:**
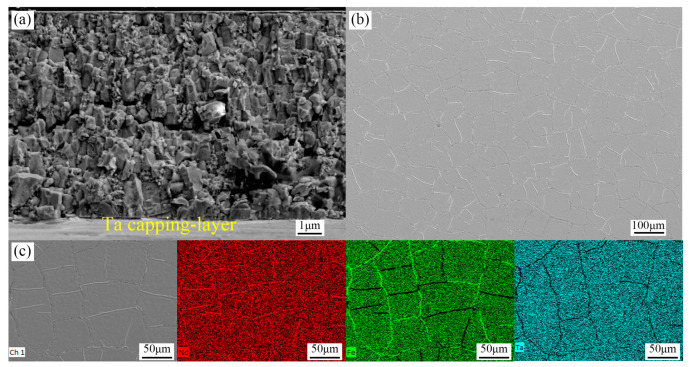
SEM images of Si(100)/Ta(100 nm)/Nd_16.4_Fe_74.6_B_9_(9 μm)/Ta(100 nm) film (**a**) cross-sectional morphology; (**b**) surface topography; and (**c**) distribution of surface EDS elements.

**Figure 7 materials-16-04071-f007:**
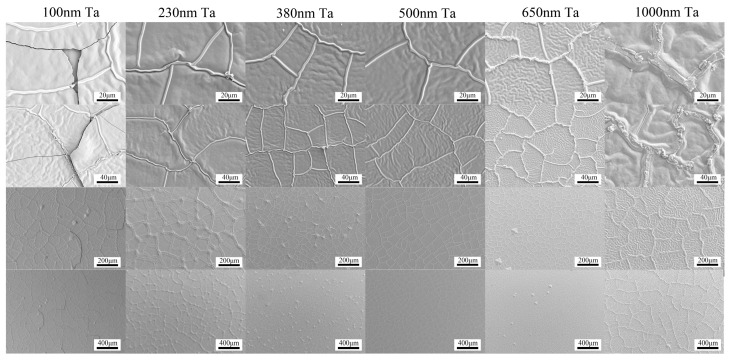
The surface SEM images of Si (100)/Ta (x nm)/Nd_14.5_Fe_74.5_B_11_ (6 μm)/Ta(100 nm) (x = 100, 230, 380, 500, 650, 1000) films after annealing at 850 °C for 20 min.

**Figure 8 materials-16-04071-f008:**
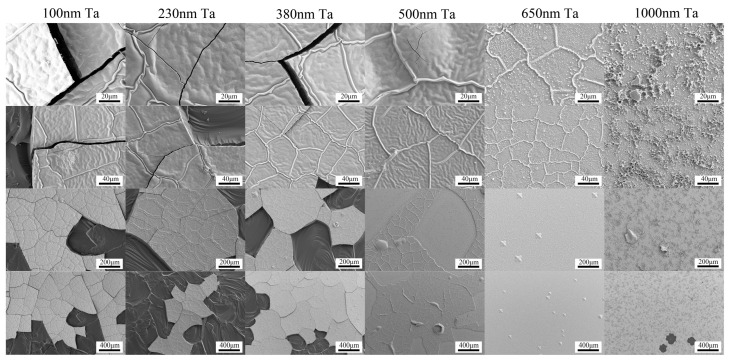
The surface SEM images of Si (100)/Ta (x nm)/Nd_14.5_Fe_74.5_B_11_ (6 μm)/Ta(100 nm) (x = 100, 230, 380, 500, 650, 1000) films after annealing at 900 °C for 20 min.

**Figure 9 materials-16-04071-f009:**
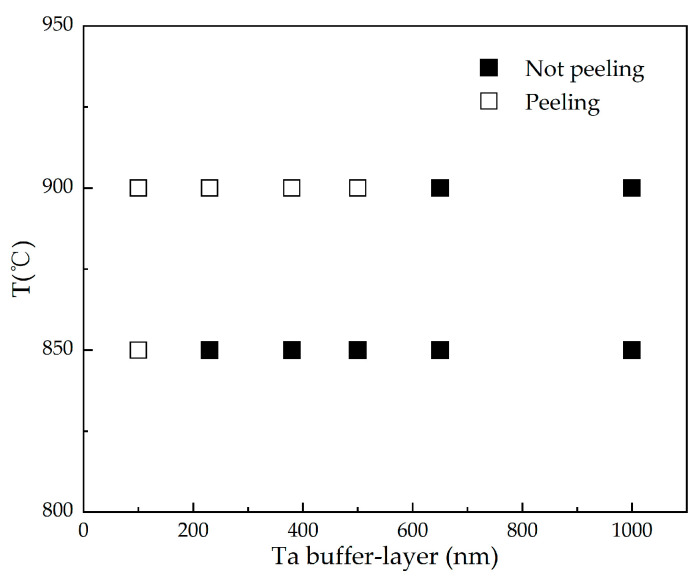
Peeling behaviors of NdFeB films with different Ta buffer layer thicknesses at different annealing temperatures.

## Data Availability

The data presented in this study are available on request from the corresponding author.

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
