# Peer review of "Achievement of High Perpendicular Anisotropy and Modification of Heat Treatment Peeling in Micron-Thickness Nd-Fe-B Films Used for Magnetic MEMS"

_materials, 2023, doi:10.3390/ma16114071_

Round 1
Reviewer 1 Report
The structure of the paper is rather simple and easy to follow and the aim of the authors was clearly stated. The tendency to peel is not quantified. Instead, the reader the reader is expected to judge the peeling proneness from the SEM images, which are reasonably convincing in my opinion. Nevertheless, I wonder if a test similar to the “Scotch-tape test” could be used to see what percentage of the film comes off the sticky surface of the tape is ripped off from the film. However, the images of Fig.8 gives some information along these lines, as large flakes are missing in the films covered by only a thin Ta-layer.
As XRD patterns were recorded for the films and the c-axis orientation is relevant for the magnetic anisotropy it could have been interesting to fit the XRD data with a model taking into account the texturizing. If is difficult to perform full Rietveld analysis one could perhaps fix most of the parameters before attempting to fit the degree of texturizing.
However, this and the above mentioned analysis of the degree of peeling is not something I demand, as I think the manuscript can be published almost as is.
L. 59 textured is misspelled
L..68 ...exceeds the critical cracking stress of the film....
L. 72 ... a mask....
L.74 ... the film fabrication more complicated...
L. 78 ... annealing with a temperature gradient could help to....
L. 79 A 9 um thick NdFeB film....
L. 82 ... of the NdFeB film...
L. 86 A commercial target....
L. 88-89 ... conditions of a .... temperature of 300... a DC sputtering power of...., and an Ar pressure of....
L. 94 I am not very fond of the sccm unit, perhaps cm³/min at STP
L. 96 An X-ray....
L. 173 ...latter....
L. 272. replace 'comfortingly' with some better word scientific word
L. 279 Increasing the .... of the Ta buffer...
Author Response
Dear Reviewer,
We thank the reviewers for the efforts given in reading our manuscript (materials-2384442) and having taken the time to provide valuable comments. We have revised the manuscript according to your requirements carefully. The comments, our responses and the main modifications are listed as follows:
Reviewer #1
The structure of the paper is rather simple and easy to follow and the aim of the authors was clearly stated. The tendency to peel is not quantified. Instead, the reader the reader is expected to judge the peeling proneness from the SEM images, which are reasonably convincing in my opinion. Nevertheless, I wonder if a test similar to the “Scotch-tape test” could be used to see what percentage of the film comes off the sticky surface of the tape is ripped off from the film. However, the images of Fig.8 gives some information along these lines, as large flakes are missing in the films covered by only a thin Ta-layer.
As XRD patterns were recorded for the films and the c-axis orientation is relevant for the magnetic anisotropy it could have been interesting to fit the XRD data with a model taking into account the texturizing. If is difficult to perform full Rietveld analysis one could perhaps fix most of the parameters before attempting to fit the degree of texturizing.
However, this and the above mentioned analysis of the degree of peeling is not something I demand, as I think the manuscript can be published almost as is..
1 Comment 1: L. 59 textured is misspelled
Response: Thank you for your suggestion. The spelling mistake of ‘textureed’ has been revised in the line 65 of page 2.
2 Comment 2: L.68 ...exceeds the critical cracking stress of the film…
Response: Thank you for your suggestion. The sentence " During thermal annealing, the thermal stress on the film exceeds films critical cracking stress, making it easy to peel off the substrate " has been changed into “During thermal annealing, the thermal stress on the film exceeds the critical cracking stress of the film, making it easy to peel off the substrate” (lines 74-75)
3 Comment 3: L. 72 ... a mask…
Response: Based on the reviewer’s advice, the sentence " Studies have shown that the thermal stress on the film peeling during thermal annealing can be weakened by aetching the mask trench on the Si substrate and then depositing the NdFeB film [8,26]." has been changed into “Studies have shown that the thermal stress on the film peeling during thermal annealing can be weakened by aetching a mask trench on the Si substrate and then depositing the NdFeB film [11,29].” (lines 77-79)
4 Comment 4: L.74 ... the film fabrication more complicated…
Response: Based on the reviewer’s advice, the word “become” is deleted from the sentence “However, this process makes the film fabrication become more complicated.” (lines 79-80)
5 Comment 5: L. 78 ... annealing with a temperature gradient could help to...
Response: Based on the reviewer’s advice, the sentence "It is found that annealing with temperature gradient could help improve the texture of thick films.” has been changed into “It is found that annealing with a temperature gradient could help to improve the texture of thick films.” (lines 84-86)
6 Comment 6: L. 79 A 9 um thick NdFeB film....
Response: Based on the reviewer’s advice, change the word "The" to "A" (line 85)
7 Comment 7: L. 82 ... of the NdFeB film....
Response: Thank you for your suggestion. The sentence “and it was found that increasing the Ta buffer thickness can effectively prevent the peeling of NdFeB film from the substrate during the high-temperature annealing.” has been changed into “and it was found that increasing the Ta buffer thickness can effectively prevent the peeling of the NdFeB film from the substrate during the high-temperature annealing.” (lines 87-89)
8 Comment 8: L. 86 A commercial target...
Response: Based on the reviewer’s advice, change the word "The" to "A" (line 92)
9 Comment 9: L. 88-89 ... conditions of a .... temperature of 300... a DC sputtering power of...., and an Ar pressure of....
Response: Based on the reviewer’s advice, the sentence “and the deposition of Ta is completed under the conditions of deposition temperature 300°C, DC sputtering power 120W, Ar pressure 0.8Pa, respectively.” has been changed into “and the deposition of Ta is completed under the conditions of a deposition temperature 300°C, a DC sputtering power 120W, and an Ar pressure of 0.8Pa, respectively.” (lines 93-95)
10 Comment 10: L. 94 I am not very fond of the sccm unit, perhaps cm³/min at STP.
Response: Thank you for your suggestion. The incorrect gas flow unit “sccm” has been corrected to “cm3 min−1” (line 101)
11 Comment 11: L. 96 An X-ray...
Response: Based on the reviewer’s advice, change the word "The" to "An" (line 103)
12 Comment 12: L. 173 ...latter....
Response: Based on the reviewer’s advice, change the word "later" to "latter" (line 180)
13 Comment 13: L. 272. replace 'comfortingly' with some better word scientific word.
Response: Thank you for your suggestion. change the word " comfortingly " to "noteworthy" (line 281)
14 Comment 14: L. 279 Increasing the .... of the Ta buffer...
Response: Based on the reviewer’s advice, the sentence “The increasing the thickness of Ta buffer-layer is beneficial for inhibiting the peeling of micrometer-scale films during high temparture annealing.” has been changed into “Increasing the thickness of the Ta buffer-layer is beneficial for inhibiting the peeling of micrometer-scale films during high temparture annealing.” (lines 288-290)
Reviewer 2 Report
In this manuscript the authors present a study on how to improve the film quality of NdFeB films with out of plane moment orientation for film thicknesses larger than 2 microns. It is investigated how the magnetic anisotropy can be preserved even for large thickness samples as well as how to avoid peeling off from the substrate which becomes a problem for thick films.
The comparison of films with different Nd concentrations shows a dependence of magnetic properties on the Nd content and a reduction of anisotropy with increasing Nd content. The XRD patterns support a maximum anisotropy with only c-axis reflections for the lowest Nd content, whereas other reflections appear for larger Nd concentration.
The comparison of hysteresis loops for different thickness films between 2 and 9 microns after gradient annealing show the presence of a strong anisotropy for all thicknesses, which is attributed to the effect of the gradient annealing. Also, XRD patterns show that thick gradient annealed films will not lose the desired anisotropy with the increased thickness.
It is found that increasing the Ta buffer layer avoids peeling off from the Si substrate during the annealing process. The buffer layer needs to be increased further for higher annealing temperatures.
In summary, the manuscript presents a careful and in-depth study of the behavior and properties of Nd-Fe-B films on a Si substrate for increased film thicknesses and shows a way to reliably produce film thicknesses with out of plane anisotropy. I recommend acceptance for publication in Materials.
Please consider the following comments:
· MEMS should be defined in main text, not in abstract.
· Mr and Ms should be defined.
Author Response
Dear Reviewer,
We thank the reviewers for the efforts given in reading our manuscript (materials-2384442) and having taken the time to provide valuable comments. We have revised the manuscript according to your requirements carefully. The comments, our responses and the main modifications are listed as follows:
1 Comment 1: MEMS should be defined in main text, not in abstract.
Response: Thank you for your suggestion. We added the definition of MEMS (line 47)
2 Comment 2: Mr and Ms should be defined.
Response: Thank you for your suggestion. “Mr” be defined “remanence” (line 126). “Ms” be defined “saturation magnetization strength” (lines 126-127).
Reviewer 3 Report
The scientific approach is well structured and solidly experimentally founded. The proposed manuscript organization is logically conducted, and the magnetic films sample measurements are compelling for the proposed approach.
A couple of remarks can be made:
1.At page 2, line 49 it must be the remanence (Br) instead of coercivity (“Br=1.6T” not (“Hc=1.6T”).
2. At page 3, line 123, I would suggest indicating explicitly the sputtering power of 140 W, as mentioned in the Figure 1 caption.
3. As a general remark, either in the Introduction or Conclusions sections, I would suggest that more information regarding the practical applications of the MEMS containing Nd-Fe-B permanent magnetic films should be presented. Additionally, in relation to these applications, to justify and discuss the importance of the magnetic films behavior and characteristics improvement achieved through the research reported within the proposed manuscript.
The English text quality is excellent, both in what grammar and style are concerned. A couple of minimal corrections are nonetheless necessary:
1. At page 2, line 68 correct "films" to "the film's".
2. At page 3, line 130 correct "has" to "have".
Author Response
Dear Reviewer,
We thank the reviewers for the efforts given in reading our manuscript (materials-2384442) and having taken the time to provide valuable comments. We have revised the manuscript according to your requirements carefully. The comments, our responses and the main modifications are listed as follows:
1 Comment 1: At page 2, line 49 it must be the remanence (Br) instead of coercivity (“Br=1.6T” not (“Hc=1.6T”).
Response: Thank you for your suggestion. We reconfirmed the data in the literature that the film reported by Dempsey et al. et al. does have a coercivity of 1.6T, while the remanence of 1.4T. We have supplemented the manuscript with remanence data. Manuscript revisions are shown below:
“Dempsey et al. reported the fabrication of 5 μm thick NdFeB films with strong perpendicular anisotropy on Si wafer (((BH)max=400kJ/m3, μ0Hc=1.6T, μ0Mr=1.4T) [8].” (lines 54-55)
- Dempsey N M, Walther A, May F, et al. High performance hard magnetic NdFeB thick films for integration into mi-cro-electro-mechanical systems[J]. Applied physics letters, 2007, 90(9): 092509.
2 Comment 2: At page 3, line 123, I would suggest indicating explicitly the sputtering power of 140 W, as mentioned in the Figure 1 caption.
Response: Based on the reviewer’s advice, the sentence “Under the same sputtering power, as the Nd content of the target increases, the Nd content in the film also increases, resulting in an increase in the coercivity.” has been changed into “Under 140W sputtering power, as the Nd content of the target increases, the Nd content in the film also increases, resulting in an increase in the coercivity.” (lines 130-132)
3 Comment 3: As a general remark, either in the Introduction or Conclusions sections, I would suggest that more information regarding the practical applications of the MEMS containing Nd-Fe-B permanent magnetic films should be presented. Additionally, in relation to these applications, to justify and discuss the importance of the magnetic films behavior and characteristics improvement achieved through the research reported within the proposed manuscript.
Response: Thank you for your suggestion. We have supplemented the information on practical applications of MEMS containing Nd-Fe-B permanent magnet films. Add the following sentence to the conclusions:
“Permanent magnets are used in magnetic microelectromechanical systems (MEMS) through electrostatic, electromagnetic, and other effects, typically ranging in thickness from a few microns to a few millimeters. High-performance permanent magnetic films are the key to implementing magnetic MEMS components [1-3].” (lines 39-42)
- Dempsey N M. Hard magnetic materials for MEMS applications[J]. Nanoscale Magnetic Materials and Applications, 2009: 661-683.
- Cugat O, Delamare J, Reyne G. Magnetic microsystems: mag-MEMS[M]. Springer Netherlands, 2008.
- Arnold D P, Wang N. Permanent magnets for MEMS[J]. Journal of microelectromechanical systems, 2009, 18(6): 1255-1266.
“Micromagnetic components made of high perpendicular anisotropic Nd-Fe-B permanent magnet films integrated into semiconductor Si can be applied to mi-cromotors, switches, brakes, energy harvesters, etc. Our findings contribute to the devel-opment of these devices.” (lines 294-297)
4 Comment 4: At page 2, line 68 correct "films" to "the film's".
Response: Based on the suggestions of the two reviewers, the sentence " During thermal annealing, the thermal stress on the film exceeds films critical cracking stress, making it easy to peel off the substrate " has been changed into “During thermal annealing, the thermal stress on the film exceeds the critical cracking stress of the film, making it easy to peel off the substrate” (lines 74-75)
5 Comment 5: At page 3, line 130 correct "has" to "have".
Response: Based on the reviewer’s advice, change the word "hsa" to "have" (line 135)